**A morpho-tectonic approach to the study of earthquakes in Rome**
Fabrizio Marra[1]*, Alberto Frepoli[1], Dario Gioia[2], Marcello Schiattarella[3], Andrea
Tertulliani[1], Monica Bini[4], Gaetano De Luca[1], Marco Luppichini[5]
[1]Istituto Nazionale di Geofisica e Vulcanologia, Via di Vigna Murata 605, 00143 Rome, Italy
[2]Istituto di Scienze del Patrimonio Culturale (ISPC), Consiglio Nazionale delle Ricerche, Tito Scalo,
I-85050 Potenza, Italy
[3]Dipartimento delle Culture Europee e del Mediterraneo (DiCEM), Università degli Studi della
Basilicata, I-75100 Matera, Italy; marcello.schiattarella@unibas.it
[4]Dipartimento di Scienze della Terra, Università di Pisa, Italy
[5]Dipartimento di Scienze della Terra, Università di Firenze, Italy
*corresponding author: fabrizio.marra@ingv.it
**Abstract**
Rome has the world's longest historical record of felt earthquakes, with more
than 100 events during the last 2,600 years. However, no destructive earthquake
has been reported in the sources and all of the greatest damage suffered in the
past has been attributed to far-field events. While this fact suggests that a
moderate seismotectonic regime characterizes the Rome area, no study has
provided a comprehensive explanation for the lack of strong earthquakes in the
region. Through the analysis of the focal mechanism and the morphostructural
setting of the epicentral area of a "typical" moderate earthquake (ML=3.3) that
recently occurred in the northern urban area of Rome, we demonstrate that this
event reactivated a buried segment of an ancient fault generated under both a
different and a stronger tectonic regime than that which is presently active. We
also show that the evident structural control over the drainage network in
this area reflects an extreme degree of fragmentation of a set of buried faults
generated under two competing stress fields throughout the Pleistocene. Small
faults and a present-day weaker tectonic regime with respect to that acting
during the Pleistocene might explain the lack of strong seismicity in the long
historical record, suggesting that a large earthquake is not likely to occur.

**Key words:** Rome; geomorphology; streambed analysis; structural geology; earthquakes; seismotectonics

## 1. Introduction

On May 11$^{th}$ 2020, a moderate ($M_L$=3.3, Io= IV MCS) yet broadly felt earthquake awoke most of the Rome's inhabitants at 05:03 a.m. (local time) (for details see https://e.hsit.it/24397691/index.html). While producing no damage, the shaking alarmed many citizens, who searched for information and reassurance through the dedicated informative sources such as the INGV (Italian National Institute of Geophysics and Volcanology) website. Others, instead, preferred to trust on several popular beliefs which state that "Rome couldn't be struck by a Big One" (i.e., a destructive earthquake with M>7.0), such as the mitigating effect of the catacomb voids (trivial simplification from the Aristotelian theories), or the protection granted by the Pope's presence. It is very likely that only few people based their reactions upon a learned knowledge of the actual seismotectonics features of the Rome's area. Indeed, even if a series of specialized studies have been published in the last 20 years, a dedicated paper investigating the reasons why Rome would not be affected by large earthquakes is still missing in the scientific literature. Filling this gap is the aim of the present paper in which we present a seismic study of the May 11$^{th}$ 2020 earthquake, coupled with a statistical analysis of streambed directions in the epicentral area. We identify the geometry of the seismogenic structure responsible for this M=3.3 event, and we frame it within the overall geo-morpho-structural setting of the Rome's area, providing insights on the seismo-tectonic features of this region.


## 2. Seismicity of the Rome's area


Our knowledge on the earthquakes that affected the roman area can be resumed
from the seismic catalogues' records (Guidoboni et al., 2018; Rovida et al., 2020
and from the literature (e.g., Tertulliani and Riguzzi, 1995; Molin and Rossi.,
2004; Galli and Molin 2014; Tertulliani et al., 2020) as follows:
• very few events caused significant damage in the city (1349, 1703, 1915),
according to the studies mentioned above; all these large earthquakes
occurred in the Apennines mountain range;
• some other seismogenic areas surrounding Rome (e.g., the Colli Albani
Volcanic District) generated events that caused moderate damage;
• the Province of Rome (hereafter GAR, is the present metropolitan area of
Rome) is periodically affected by low to moderate magnitude local
earthquakes which is not supposed to cause significant damage.
• Uncertain events. Catalogue records quote several earthquakes that
provoked some damage in Rome (see table 1). Most of such events,
occurred during the Roman Age and Early Middle Ages are poorly
documented and therefore not localizable.
A summary of the historical and instrumental seismicity of the GAR is shown in
Figure 1. Evidently, the completeness of our knowledge of seismicity decreases
going back in time. In the period of ancient Rome, as well in the Early Middle
Ages, strong earthquakes would seem hit Rome, sometime causing damage,
whose origin is still unknown. The difficulty to understand if such earthquakes
were generated by local or far-field sources depends on the documentary
accounts: the earthquake was considered a prodigy, and as such, interpreted as a
divine foretelling. Information on effects, damage or victims was often neglected,
and very rarely documented. For these reasons we are not able to distinguish
with reliability if such ancient events were originated, for example, in the
Apennines region, or near Rome (in italic in table 1). In table 1 the earthquakes
that hit Rome with a local intensity greater than 6 are listed.

| Int. in Rome | Year | Epicentral Area | Epic Int Io | Mw |
|---|---|---|---|---|
| *7-8* | *83 BC* | *Rome* | *7-8* | *5.4* |

| | | | | |
|---|---|---|---|---|
| *7-8* | *72 BC* | *Rome* | *7-8* | *5.4* |
| *7-8* | *15* | *Rome* | *7-8* | *5.4* |
| *8* | *51* | *Rome* | *8* | *5.6* |
| *8* | *443* | *Rome* | *8* | *5.6* |
| *7-8* | *484* | *Rome* | *7-8* | *5.4* |
| *7-8* | *801* | *Rome* | *7-8* | *5.4* |
| *7* | *1091* | *Rome* | *7* | *5.1* |
| 7-8 | 1349 | Central Apennines | 9 | 6.3 |
| 5-6 | 1703 | Central Apennines | 11 | 6.9 |
| 6 | 1703 | Central Apennines | 10 | 6.7 |
| 6 | 1730 | Central Apennines | 9 | 6.0 |
| 6-7 | 1812 | Rome | 6-7 | 4.9 |
| 5-6 | 1895 | Rome | 6-7 | 4.8 |
| 6-7 | 1899 | Albani Hills | 7 | 5.1 |
| 6-7 | 1915 | Central Apennines | 11 | 7.1 |
| 6 | 1927 | Albani Hills | 7 | 4.9 |

Table 1. List of earthquakes that caused documented damage in the present GAR.
The oldest events (italic in table) are not constrainable. (Data from Guidoboni et
al., 2018; Rovida et al., 2021; Tertulliani et al-; 2020).

It is interesting to note, from the seismic hazard point of view, that the epicenter
of several more constrainable historical events, that occurred in the Roman
countryside, are nowadays included in the GAR territory, densely urbanized.
Within this limited territory we can anyway discriminate some different clusters
of seismicity, in particular SE and NE of the City center. Of the first cluster are
part the 1812, 1895, 1995 earthquakes, while the 1901 and 2020 events are
located NE of the city (Figure 1). Very likely this seismicity feature is due to the
activity of different seismotectonic structures.

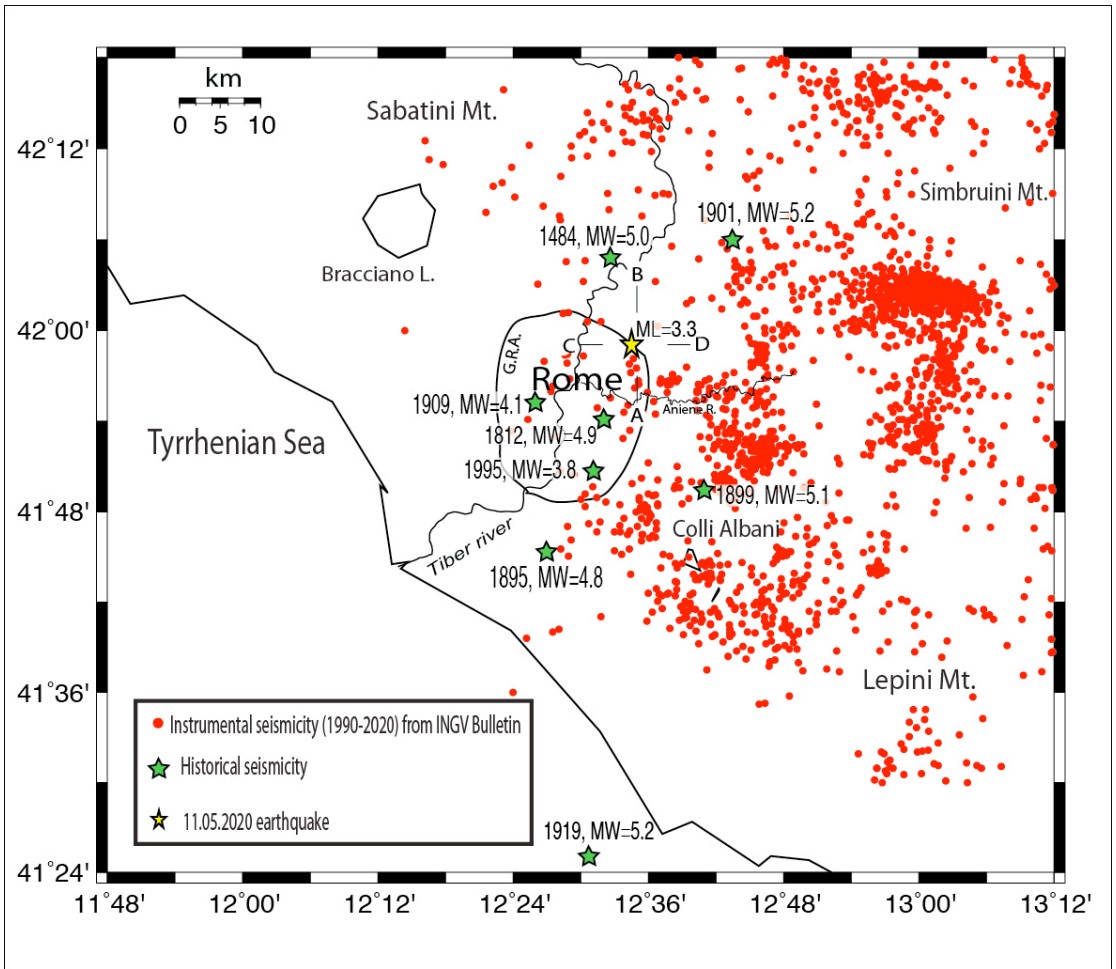


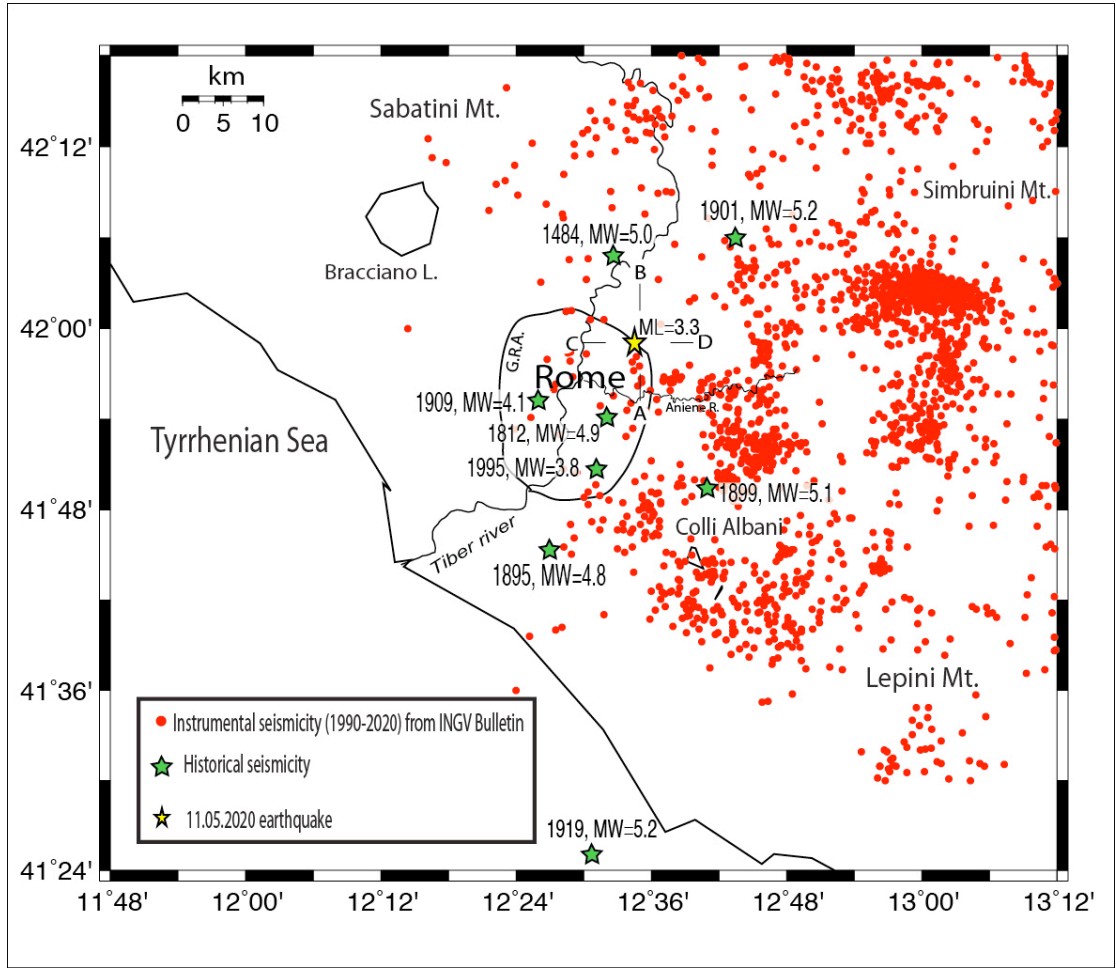

**Figure 1.** Map showing the seismicity of the Rome's area and mainshock location (blue star) of the 11.05.2020 earthquake. A-B and C-D are the  cross-sections in Figure 4b. G.R.A. is the beltway around Rome.

## 3. Regional tectonic setting

In approaching the geodynamics of this region the contribution of three main mechanisms of deformation should be considered, as proposed in Faccenna et al. (1996):

i) the NW-SE shortening (arrow #1 in Figure 2b) induced by the convergence of Africa and Europe (Tapponier, 1977);

ii) the sinking of the Ionian slab (arrow #2 in Figure 2b), producing the eastward migration (arrow #3) of the Apennine arc, and consequent back-arc extension (arrow #4) in the Tyrrhenian region (Malinverno and Ryan, 1986; Patacca and Scandone, 1989);

iii) the gravitational spreading of the overthickened crust (arrow #5 in Figure 2b) in the Apennine crustal wedge (Reutter et al., 1980; Horvath and Berckhemer, 1982).

All these mechanisms are to be considered presently active in the Northern
Apenninic arc on the basis of seismic and stress-field indications (Selvaggi and
Amato, 1992; Amato et al., 1993; Frepoli and Amato, 1997; Mariucci et al., 1999;
Lucente and Speranza, 2001; Montone and Mariucci, 2016). Moreover, crustal
thinning induced by extension was coupled with asthenospheric bulging (arrows
#6 in Figure 2b), leading to the back-arc volcanism on the Tyrrhenian margin
(Serri, 1997, and references therein). Such phenomena, and related magma
underplating, enhanced the extensional processes (arrow #6′ in Figure 2b) in a
feedback mechanism in this region. In this regard, it is fundamental to notice that
the Rome area and the Alban Hills are at the southeastern margin of the Latium
Magmatic Province (Serri et al., 1993), and that very scanty volcanic activity
occurred in the area between Rome and the Ortona-Roccamonfina Line (O-R in
Figure 2a), which is considered (Patacca et al., 1990) a major geodynamic
boundary separating the Central and Southern Apennines (Figure 2a). According
to Marra (1999, 2001), the Sabina shear zone (Alfonsi et al., 1991) represents the
northern boundary of this crustal disengagement zone. Based on its proximity to
the Sabina shear zone, and in agreement with the numerous field evidence of
fault kinematics (Faccenna et al., 1994a, 1994b; Marra, 2001; Marra et al., 2004)
and the peculiar eruptive behaviour of the Alban Hills Volcanic District (Marra et
al., 2009), Frepoli et al. (2010) proposed that the transpressional stress regime
has been the prevailing one in this region during Quaternary times, and that it is
temporarily superimposed by the extensional regime during periods of incoming
volcanic activity and/or increased extensional activity (depending on which is to
be considered cause and which effect) on the Tyrrhenian margin (Figure 2b).


**4. Morpho-structural features of the Rome's area**
The morpho-structural setting of the Roman area originates in the deformation
of the geological substrate by combined faulting processes and erosion of rivers
and streams (Del Monte et al., 2016). Although partially obliterated by millennia
of anthropic interventions, it presents some evident and peculiar traits, whose
analysis allows us to understand the features of the tectonic forces (and related
stress-fields) that acted in the geological past through present time (Marra,
2001) (Fig. 2). Such analysis also consents to interpret the origin of the
earthquakes that nowadays affect this area.

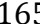

**Figure 2.** Structural scheme of central Italy showing the competing tectonic force fields and the
main faults associated with them that acted in the Middle-Upper Pleistocene. See text for
comments and explanations.


If we could see what the topography was like before the foundation of the City,
the area of Rome would appear as a large flat sector, deeply engraved and
dissected by the valleys of the tributary streams of the Tiber and Aniene Rivers,
and by the wider ones of the two main watercourses. While these features are
less visible in the historical center of Rome, they are still well recognizable
through a Digital Elevation Model (DEM) in its surrounding territory, as
highlighted in Fig. 3.

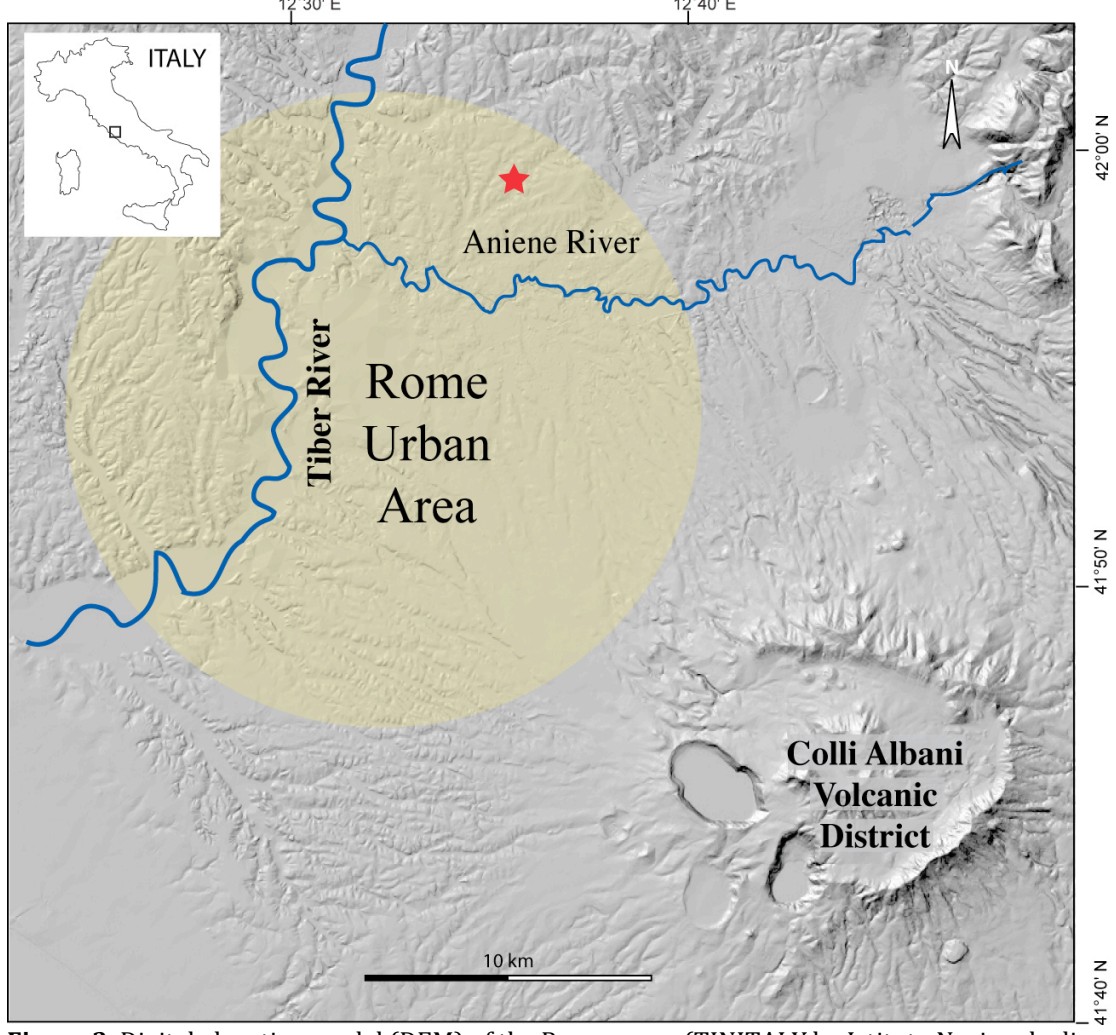

**Figure 3.** Digital elevation model (DEM) of the Roman area (TINITALY by Istituto Nazionale di
Geofisica e Vulcanologia (INGV), published with a CC BY 4.0 license; available at:
https://doi.org/10.13127/TINITALY/1.0), showing the strongly marked characters of the river
and stream incisions that form the hydrographic network afferent to the Tiber and the Aniene
Rivers. Location of the 10.05.2020 earthquake is also shown (red star).

Most of the tabular surface highlighted by the shaded area in Fig. 3 is a
"pyroclastic plateau" created by the emplacement of large coulters of volcanic
deposits erupted from the Colli Albani and Monti Sabatini districts. These are
represented by pyroclastic flows, originated by the collapse of the sustained
eruptive column, and air-fall products such as windblown pumice, scoria and
lapilli. The deposition of these volcanic products, starting from around 600,000
years ago (Marra et al., 2014; Gaeta et al., 2016), leveled the ground creating a
thick, layered blanket of sediments which was soon after etched by the erosive
action of the watercourses. The latter, however, did not settle at random, but
progressively shifted in correspondence with embryonic fractures and fault lines
created by active tectonic deformation. The same fracturing and faulting
associated with the extensional tectonic regime which shaped the Tyrrhenian
Sea margin of central Italy during the Pleistocene allowed the magma residing in
the mantle to rise to the surface (e.g., Locardi et al., 1977, Acocella and Funiciello,
2006), originating the volcanoes of the so-called "Roman Province" (Peccerillo,
2017) (Fig. 2). An intense seismotectonic regime must have been associated to
these large extensional faults, likely producing strong earthquakes throughout
this region.

From the end of the Middle Pleistocene (125,000 years ago), the tectonic activity
began to decrease in intensity, paralleling the decrease in volcanic activity
(Marra et al., 2004a). Hence the seismogenic potential of the faults associated
with this tectonic regime must also have decreased significantly. This is one of
the reasons why Rome is today a low seismicity area. Moderate earthquakes
(M≤5.0) (Tertulliani and Riguzzi, 1995; Basili et al., 1995) are almost exclusively
concentrated in the volcanic area of Colli Albani (Amato and Chiarabba, 1995),
which is in a quiescent status (Trasatti et al., 2018). The moderate seismicity of
the Roman area reflects an active stress-field of the same nature, but weaker,
than the extensive tectonic regime that characterized the Tyrrhenian Sea margin
of central Italy for the entire Pleistocene, as revealed by the study of the focal
mechanisms of these earthquakes and borehole breakouts (Montone et al., 1995;
Montone and Mariucci, 2016). Such weaker tectonic regime, therefore,
reactivates all the faults present in this region with small movements, compatible
with their orientation with respect to the vectors of the stress-field (Frepoli et al.,
2010). The seismic events associated with this regime do not generate ground
ruptures, as it happens for strong, heavy damaging earthquakes, because the
small displacements that occur on the fault planes at depth do not propagate to
the surface. However, these movements repeated over time generate a slow and
progressive deformation of the soil, conditioning the flow direction of surface
waters, and exerting a "structural control" on the stream axes and alluvial valleys
(Marra, 2001). It follows that the hydrographic network has assumed over time a
geometry reflecting that of the faults occurring in the geological substrate.


**5. Data and Methods**
**5.1 Seismic analysis**
The small seismic sequence occurred on May 11[th] 2020 in the north-eastern area
of Rome was recorded by the Italian National Seismic Network (RSN) of the
Istituto Nazionale di Geofisica e Vulcanologia (INGV) and by the regional seismic
network of Lazio and Abruzzo (RSA) (De Luca et al., 2009; Frepoli et al., 2017)
(Figure 4). Both national and regional Italian seismic networks have been
significantly extended in the last two decades through installation of new three
components, mostly broadband, stations. In addition we integrated the dataset of
this sequence with the data of the Italian strong motions network (RAN)
operated by the National Civil Protection Department and with the IESN network
(Italian Experimental Seismic Network) of Central Italy, an amateur seismic
network equipped with very good digitizers and sensors. This dense monitoring
improved in the last decade the detection and location of the seismicity in central
Italy.
To accurately relocate the seismicity, we used the Hypoellipse code (Lahr, 1989)
and a reliable 1D $V_p$ velocity model computed by the application of a genetic
algorithm (Holland, 1975; Sambridge and Gallagher, 1993). A constant value of
1.84 $V_p/V_s$ determined with the Wadati method (Chatelain, 1978) was used.

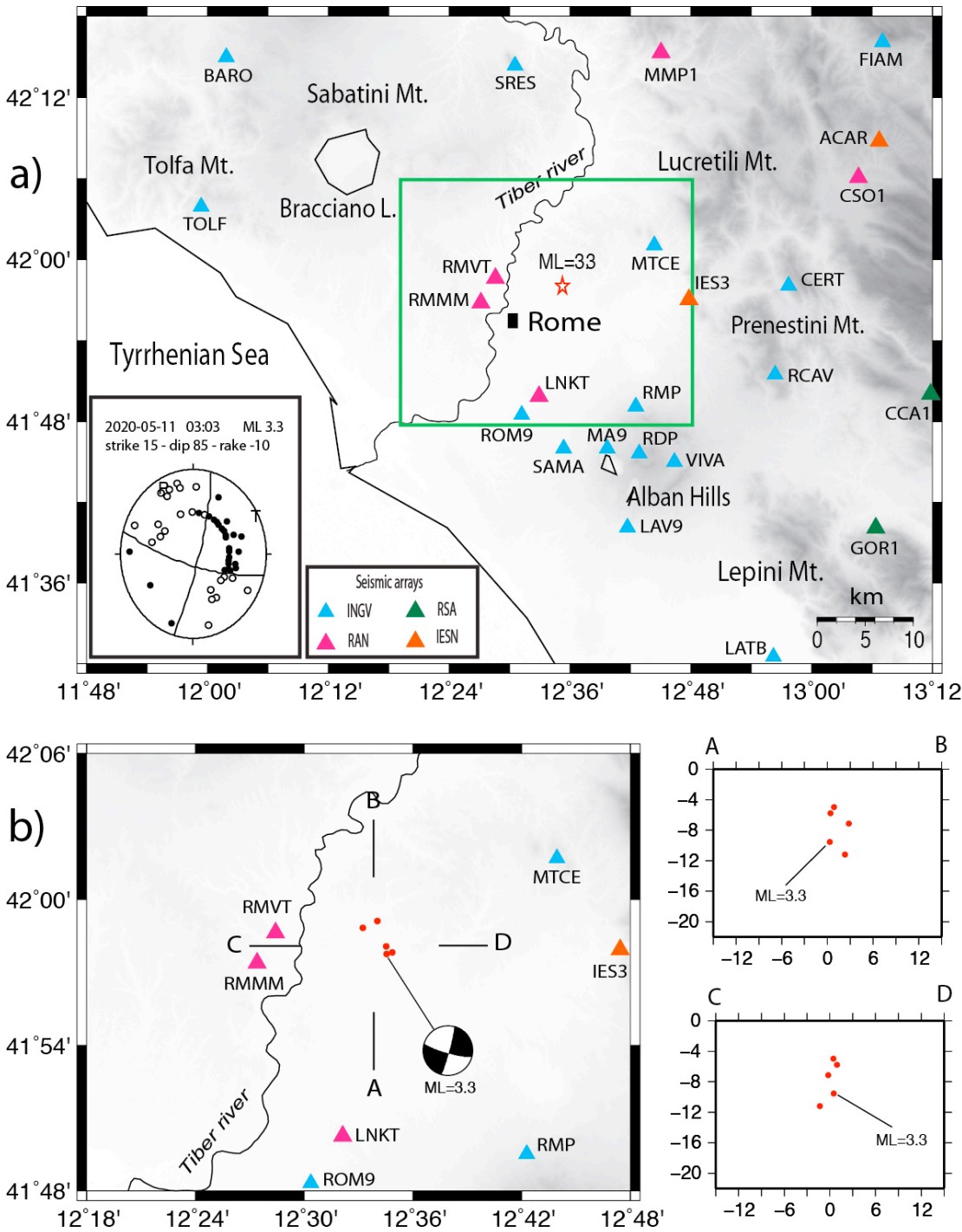


**Figure 4.** a) Distribution of the seismic stations of the Italian National Seismic Network (RSN) of
the Istituto Nazionale di Geofisica e Vulcanologia (INGV) and of the regional seismic network of
Lazio and Abruzzo (RSA) used to locate the epicenter of the 11.05.2020 event (red star). b) Map
and vertical distribution of the mainshock and two aftershocks.

256

## 5.2 Geomorphology

### 5.2.1 Previous studies

A quantitative analysis of drainage trends in the south-eastern area of Rome

bounded by the Tiber and Aniene Rivers and by the Colli Alabani volcanic district

was carried out by Marra (2001). A simple technique based on statistical analysis
of rectified directions of streambeds was applied (e.g., Ciccacci et al., 1987;
Caputo et al., 1993; Macka, 2003). Stream channel directions for the total area
and for different sectors were weighted according to three groups of length,
independent of hydrographic order, and plotted on rose diagrams.
While it is possible that rectifying drainage patterns can introduce directionality
that is unrelated to structural control, it still does indicate preferential directions
of river flow. In the case that these preferential directions of river flow were
statistically significant and different from those expected from non-structural
controls (e.g. topographic and geographic trends), they were interpreted to be
diagnostic of the structural setting. Anthropic intervention is also possible cause
of rectification of water channels, however, the linearity of the alluvial valleys
forming in the hydrographic network consents to compare and support the
directionality of the streambeds. Indeed, deep incisions and a "canyon-like"
morphology characterizes the alluvial plains forming the hydrographic network
(see Fig. 3), due to the occurrence of ca. 50 m tectonic uplift in the last 250 ka
(Marra et al., 2016).

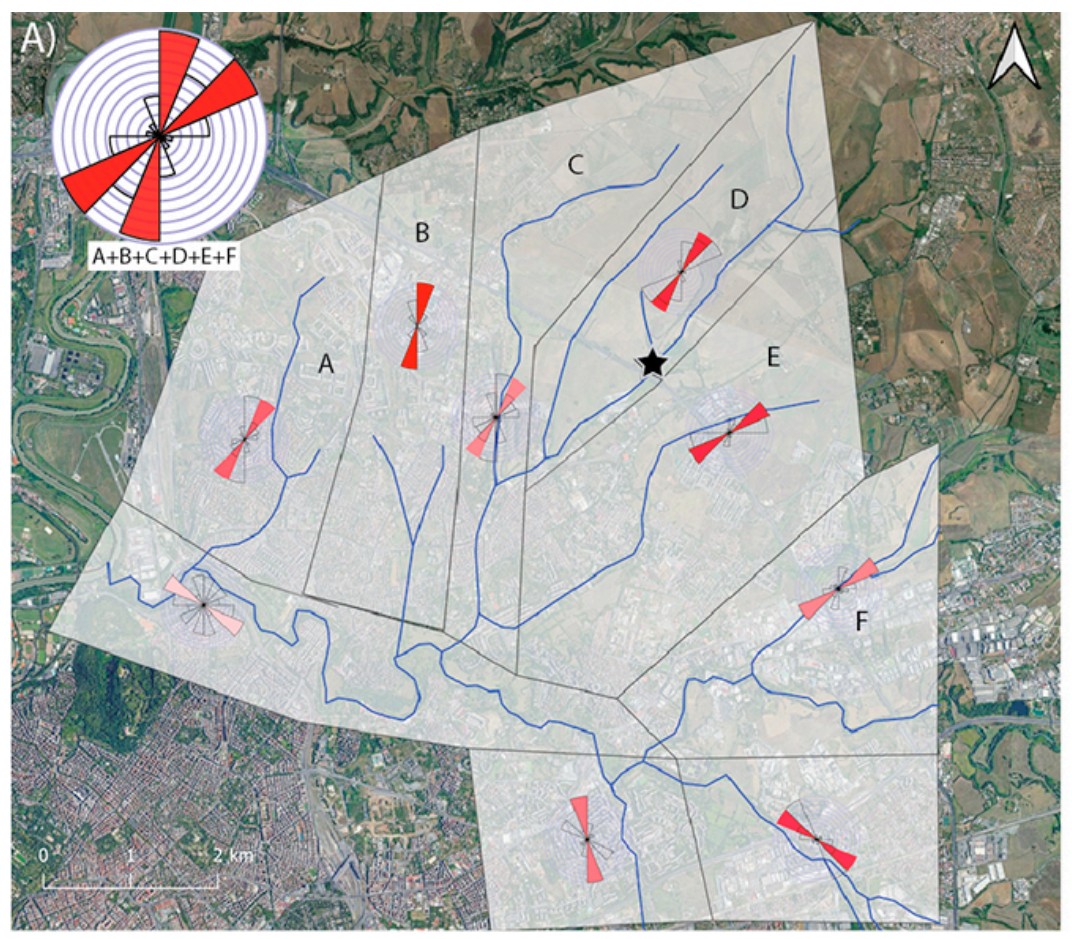

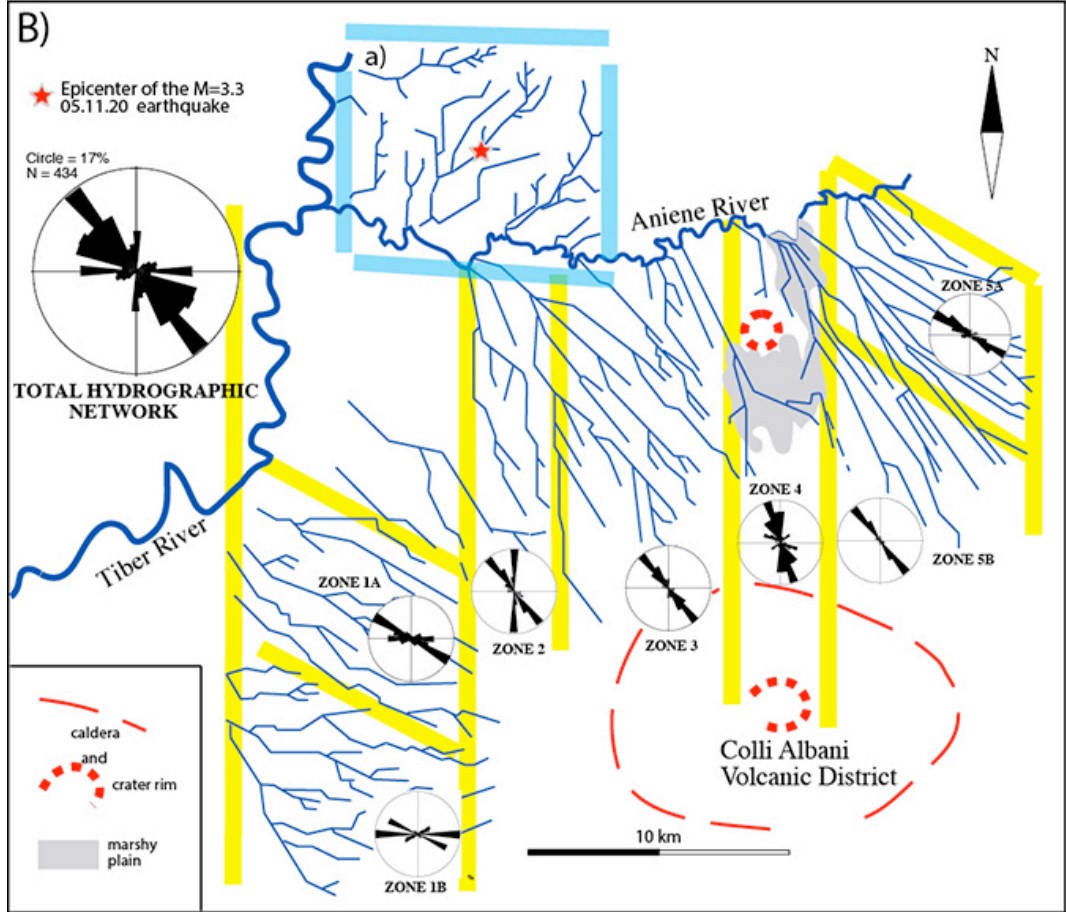

**Figure 5.** A) Result of the streambed direction analysis performed in this work
within the hydrographic basin including the epicenter area of the May 11th event
(pale-blue borders in B) is compared with that performed in the south-eastern
Roman area, between the Tiber, the Aniene and the Colli Albani (B) (Marra,
2001). Yellow lines border the different sectors of the analyzed drainage  basins.
Analysis in the historical city center was hindered by the occurrence of a
widespread anthropic cover. Basemap from Qgis QuickMapServices (available
under Creative Commons Attribution-ShareAlike 3.0 licence (CC BY-SA) at:
https://plugins.qgis.org/plugins/quick_map_services/).

Results of the analysis conducted by Marra (2001) are shown in Fig. 5B showing
that the NW-SE direction is the dominant one in the total area analysis (large
diagram in the left upper corner), as opposed to an expected radial drainage
trend descending from the Colli Albani caldera rim and affecting an
heterogeneous geologic substrate. The maximum concentration of fluvial
channel directions oriented N145° matches the strike of extension-induced faults
and fractures and agrees with the present-day stress field determined from focal
mechanisms and breakouts data in this region (Montone et al., 1995; Montone
and Mariucci, 2016). Moreover, there are significantly different concentrations in
discrete sectors delimited by the yellow lines. In particular, there are two narrow
bands (zones 2 and 4) where the N-S direction of the streambeds prevails, and
peculiar "domains" (zones 1A, 5A) where the WNW-ESE one is prevailing. The
validation of the 'tectonic' hypothesis was performed through comparison with
geometry and kinematics of fault and fractures surveyed in the area, allowing to
interpret the pattern highlighted as the result of a complex structural control in
this area, exerted by two competing stress-fields alternating each other
throughout Pleistocene times (Marra, 1999, 2001; Frepoli et al., 2010).

**5.3 Streambed analysis**
In order to compare the results with previous analysis of the regional
deformation pattern, a quantitative analysis of drainage trends has been
performed in the discrete hydrographic basin located in the sector NE of the
Tiber and Aniene Rivers confluence (Fig. 5A), within which the May 11th
earthquake occurred.
The streambed direction analysis within the hydrographic basin including the
epicenter area of the May 11th event was created by using the QGIS "Line
Direction Histogram" plugin (Tveite, 2015), that visualizes the distribution of
line segment directions as a rose diagram (weighted using the line segment
lengths). The number of bin of direction which composes the rose diagram could
be set and in this work we used 8 bins corresponding to the main cardinal
directions.  The tiles in which the area has been divided were identified
according to the main directions of streambeds.

**5.4 Drainage network anomalies and river profile analysis**
Drainage network anomalies are one of the most useful morphotectonic
indicators of active tectonics and they are widely used as an effective tool to infer
the possible control of fault activity on landscape and channels (see for example,
Boulton et al., 2014; Calzolari et al., 2016; Pavano et al., 2016; Kent et al., 2017;
Baharami, 2013). Integrated studies of possible active tectonic control on the
geometry of the drainage network frequently include analysis of river
longitudinal profiles, preferential orientation and alignments of channels, right-
angle confluences and fluvial elbows (Boulton et al., 2014; Pavano et al., 2016;
Kent et al., 2017; Gioia et al., 2018). Indeed, river profile analysis is one of the
most powerful tools for the identification of transient state of a drainage
network and recognition of knickpoints/knickzones, which represent valuable
and effective morphotectonic markers of recent crustal deformation (Whipple
and Tucker, 1999). Our approach combines the analysis of anomalies in drainage
network geometry (i.e. preferential orientation and/or alignments of channels,
fluvial elbows, right-angle confluences) with the identification of
knickpoints/knickzones of tectonic origin in transient longitudinal river profiles.
Such data have been used as morphotectonic evidences of active/recent tectonic
deformation induced by fault system responsible for the seismic activity of the
study area.
River profile analysis has been carried out according to the methods and
procedures developed by Wobus et al. (2006), Forte and Whipple (2019) using a
DEM with a spatial resolution of 10 m. Stream profile analysis is classically
carried out by identifying knickpoints or knickzones along the river longitudinal
profiles or by extracting a linear regression in a log-log slope-area graph, which
allowed us to extrapolate the concavity index (the slope of the regression) and
the steepness index (the y-intercept, that is the projection of the best-fit line that
intersects the y-axis). Knickpoints or abrupt scarps of the longitudinal profiles
can be related to tectonic- or eustatic- induced perturbations of ancient base-
levels but their formation and migration can be also related to a co-seismic fault
ruptures or deformation induced by blind faults (Kirby and Whipple, 2012). In
particular, the identification of fault-induced disturbance on channel profiles can
be performed through the recognition of linear alignments of
knickpoints/knickzones in channels with different sizes and orientations
(Boulton et al., 2014; Kirby and Whipple, 2012).
In order to investigate the possible occurrence of fault-related knickpoints and
river profile anomalies, we have investigated the river longitudinal profiles of the
main channels of the study area through the identification and mapping of
abrupt changes in river profile shape. Such data have been combined with the
morphotectonic analysis of the spatial distribution of drainage network
anomalies. Then, their spatial distribution has been used to infer the traces of
possible tectonic lineaments of the study area.

**6. Results**
**6.1 Focal mechanism and re-location of the 11 May earthquake**
The $M_L$ 3.3 mainshock (11 May at 03:03 UTC) was followed over the next two
days by only four small aftershocks with magnitude ranging from 0.7 to 1.8
(Table 2). Thanks to the high station coverage we were able to determine all
earthquake hypocenter depths with acceptable uncertainties. The average
location errors are 0.14 km (horizontally) and 0.32 km (vertically) with a
confidence level of 90%. Mainshock hypocenter is at 9.6 km of depth, while the
aftershock hypocenters are ranging from 5.0 to 11.2 km of depth (Fig. 4). The
two largest aftershocks (magnitude $M_L$ 1.8 and 1.4, respectively) have depth
between 5.0 and 5.8 km, and are located very close to the mainshock epicenter,
while the two smallest aftershocks (both magnitude $M_L$ 0.7) are located slightly
towards NW with respect to the mainshock epicenter, at 7.2 and 11.2 km of
depth. These two aftershocks are clearly unrelated with the seismogenic
structure responsible for the mainshock and are likely the effect of stress
propagation to a contiguous fault.





Table 2. List and localization parameters of the Rome sequence (May 2020).


| Date | Origin time | Lat | Lon | Depth | Azimuthal gap | RMS | Magnitude $M_L$ |
|---|---|---|---|---|---|---|---|
| 2020-05-11 | 03:03 | 41 57.77 | 12 34.54 | 9.6 | 44 | 0.14 | 3.3 |
| 2020-05-11 | 03:14.43 | 41 59.13 | 12 34.05 | 7.2 | 72 | 0.12 | 0.7 |
| 2020-05-11 | 03:14.47 | 41 58.84 | 12 33.25 | 11.2 | 73 | 0.11 | 0.7 |
| 2020-05-12 | 00:06 | 41 57.83 | 12 34.87 | 5.8 | 47 | 0.18 | 1.8 |
| 2020-05-13 | 00:07 | 41 58.08 | 12 34.53 | 5.0 | 46 | 0.20 | 1.4 |


We have computed the fault plane solution of the mainshock with the FPFIT code
(Reasenberg and Oppenheimer, 1985). First-motion polarities are 57. The focal
mechanism has a large strike-slip component (first nodal plane: strike 15, dip 85,
rake -10). T-axis is oriented in a NE-SW direction according with the general
"Antiapennine" (NE-SW) extension. Following some tectonic information of this
area, the fault plane coincides with the NNE-SSW nodal plane of the solution
which has a left-lateral strike-slip kinematics.

**6.2 Statistical analysis of streambed directions in the epicenter area**
Results of the streambed analysis in the small hydrographic basin where the
epicenter of the May 11th earthquake occurred are summarized in Fig. 5A.
The streambeds in the eastern portion of the basin (discrete sectors D, E, F)
concentrate around the NE-SW direction, which is the one expected based on the
topographic gradient, perpendicular to the Aniene River course, towards which
the catchment basin drains. In contrast, an abrupt rotation occurs in the western
portion of the basin (discrete sectors A, B, C), where the streambeds are aligned
along the NNE-SSW direction, parallel to the main watercourse of the Tiber
River. Similarly to the results obtained in the southern area by Marra (2001),
showing that the ca. N-S direction is a characteristic feature of the streambeds in
this region which is clearly independent by the geographic and topographic
control on the hydrographic network, we interpret the N-S lineaments to reflect
tectonic control on the streambeds exerted by fault activity in the analyzed basin.
As it has been remarked in previous works (e.g., Alfonsi et al., 1991; Faccenna et
al., 1994, 2008; Marra et al., 2004b) strike-slip, right-lateral N-S faults have been
active repeatedly during the Pleistocene, up to historical times. Frepoli et al.
(2010) have remarked on the direct relationship between the sectors
characterized by N-S direction of the streambeds and seismically active fault
zones. It is worth noting that the May 11th earthquake epicenter occurs on the
northern continuation of one such N-S zone (zone 2 in Fig. 5B).

**6.3 Morphotectonic analysis of the drainage network: river profile analysis**
**and drainage network anomalies**
Analysis of longitudinal river profiles of the bedrock-rivers is based on the
stream power incision model (Whipple and Tucker, 1999; Wobus et al., 2006;
Forte and Whipple, 2019) and has been carried out to evaluate the channel
response to eustatic- and tectonic-induced processes. In a first step, we prepare a
map of the normalized steepness index (ksn) with a reference concavity index of
θref = 0.45 (Fig. 6a). Ksn map allowed us to perform a preliminary analysis of the
spatial distribution of ksn values, which can be useful to individuate the sectors
of the landscape featured by knickpoints and knickzones of tectonic origin.
Moreover, a morphotectonic map showing the spatial distribution of fluvial
elbows and anomalies in drainage network geometry was also introduced (Fig.
6b). Fig. 7 shows the results of the analysis of the river profiles, which highlights
how most of the channels deviates from the typical equilibrium shape of the
longitudinal profiles. Longitudinal profiles are featured by the presence of
knickpoints and knickzones, mainly in the central reach of the river profiles.
These knickpoints appear not controlled by lithological contact and suggest a
transient state of the fluvial net induced by tectonic perturbation or eustatic
base-level variations. In particular, we detect the occurrence of convex zones or
knickpoints related to a past base-levels, as testified by the presence of a large
"terraced surfaces" at altitude ranging from 60 to 40 m a.s.l. (Fig. 7). Our analysis
also reveals the occurrence of a cluster of knickpoints in the right-orographic
side of the Aniene River with different features than the previous ones. In fact,
they can be classified as slope-break knickpoint (*sensu* Wobus et al., 2006, see
also Kirby and Whipple, 2012) and are aligned along NW-SE and N-S orientation.
Such alignments as well as the location of anomalous confluences and right-angle
elbows of the drainage network allowed us to infer the occurrence of the tectonic
lineaments mapped in Fig. 8, which can be responsible for the recent tectonic
activity that promoted the perturbation of the fluvial net.

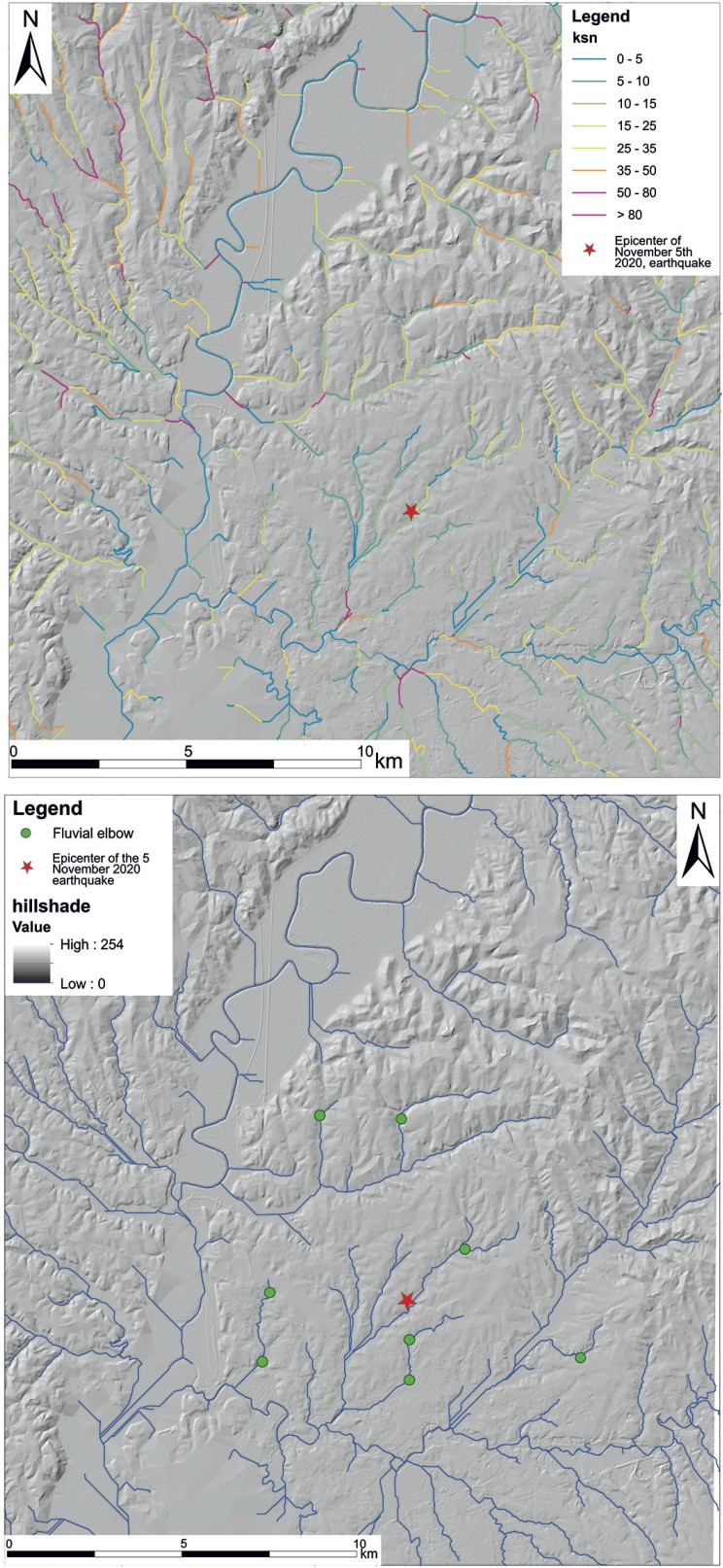


**Figure 6**. a) Hillshade of the study area and distribution of the normalized channel steepness
index (ksn, θref = 0.45); b) Drainage network of the study area and main planar anomalies of the
fluvial net. Tectonic lineaments inferred by morphotectonic analysis are also showed.

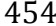

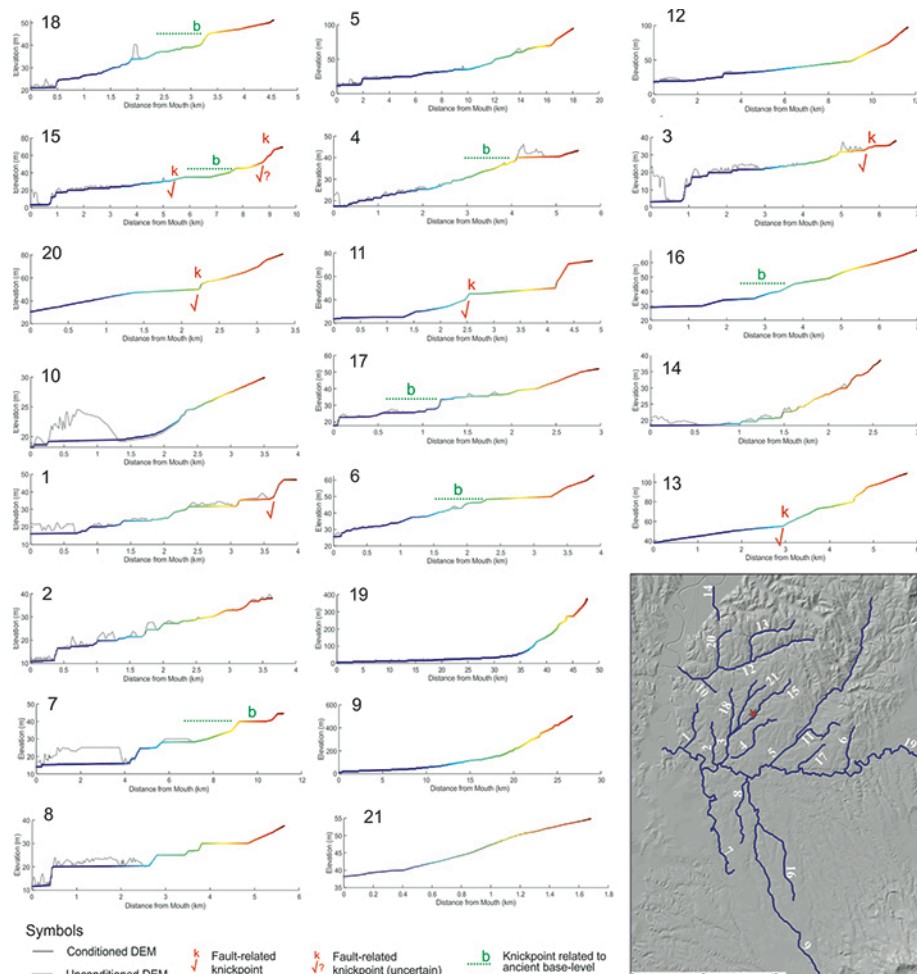

**Figure 7**. Longitudinal profiles of the main channels of the study area (location and numbering in the main map) and interpretation of the knickpoints.

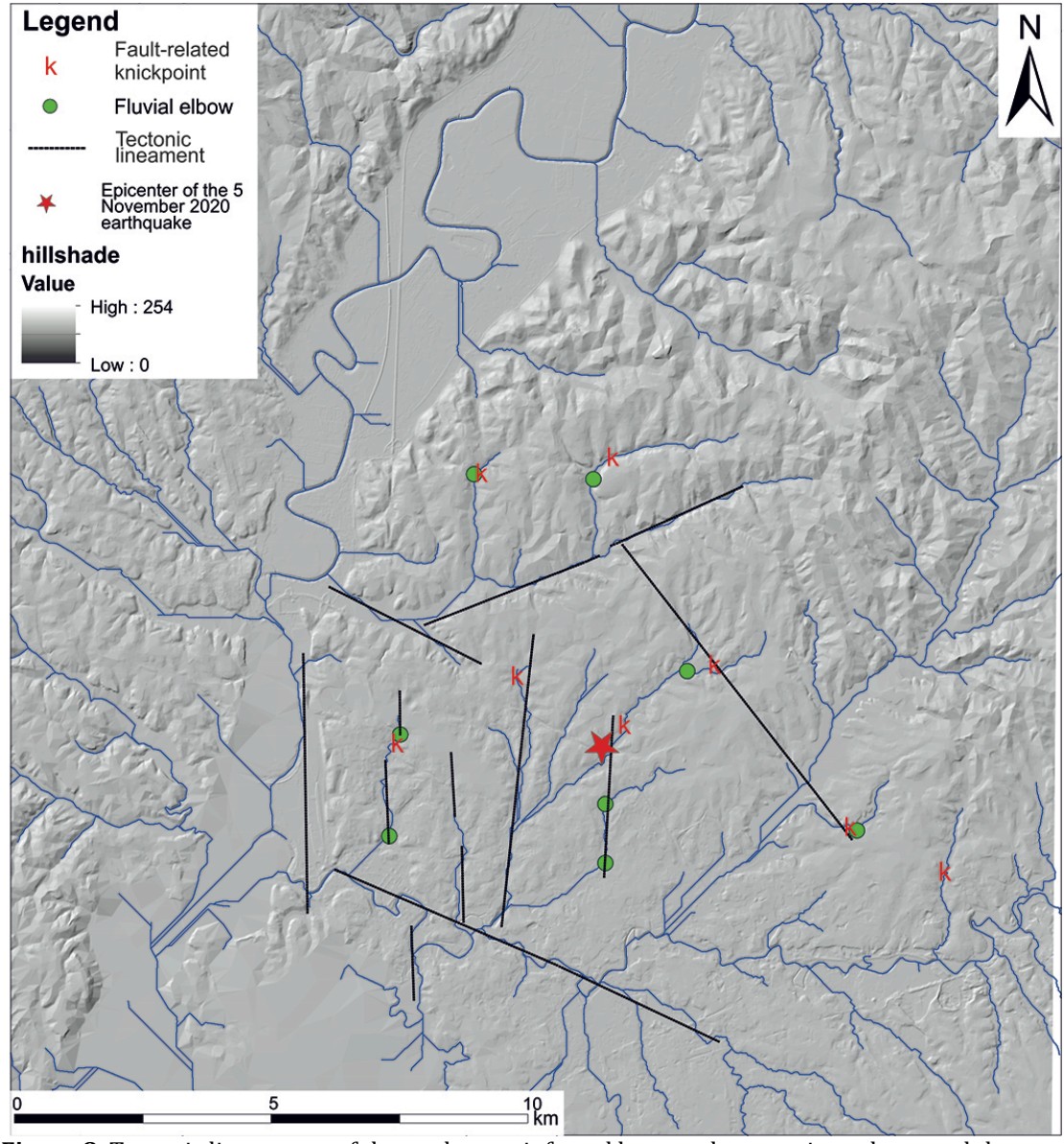

**Figure 8.** Tectonic lineaments of the study area inferred by morphotectonic analyses and the
spatial distribution of the main drainage network anomalies of the study area (i.e. fluvial elbow
and knickpoints of river profiles). Hillshade was derived by the 10 m TINITALY DEM , published
with a CC BY 4.0 license by Istituto Nazionale di Geofisica e Vulcanologia (INGV), available at:
https://doi.org/10.13127/TINITALY/1.0.
**7. Discussion**
Studies conducted during the last two decades on the geological-structural and
seismic-tectonic setting of the Roman area have shown that the geometry of the
hydrographic network reflects that of a set of buried faults (Marra, 199, 2001;
Frepoli et al., 2010). Considering the significant offsets affecting the Middle-
Pleistocene volcanic deposits in this area (e.g., Faccenna et al., 1994a, 1994b;
Marra, 2001) compared to the lack of strong events in the historical record, it is
inferred that these faults are no longer active with the seismic intensity they had
in the geological past. We conclude that they are reactivated under the effect of
the stress-field that currently acts in the upper crust and determines the genesis
of low-magnitude earthquakes in this region.
In particular, it has been shown that the drainage network pattern and the
distribution of river profile anomalies (i.e. fluvial elbows and
knickpoint/knickzones) reflect the deformation field induced on the surface by
the reactivation of these buried faults, with a set of three preferential alignments:
i-  The first displays an NW-SE "Apennine" direction, ("a" in Fig. 9), which
precisely reflects that of the large, dip-slip extensional faults that first created the
Tyrrhenian Sea marine basins (Barberi et al., 1994) and later, in the lower-
middle Pleistocene, the so-called "Tyrrhenian margin" (Fig. 2). This is a wide
hilly or sub-flat area between the Apennine chain and the present coast,
originated by the fault displacement and the "staircase" lowering of the
mountain relief (Parotto and Praturlon, 1975). The direction of these faults also
reflects the alignment of the volcanoes that developed in the Middle Pleistocene
along the Tyrrhenian margin, following the rise of magmas mainly along the
fractures in the earth's crust created by these tectonic structures (Locardi et al.,

494   1977).


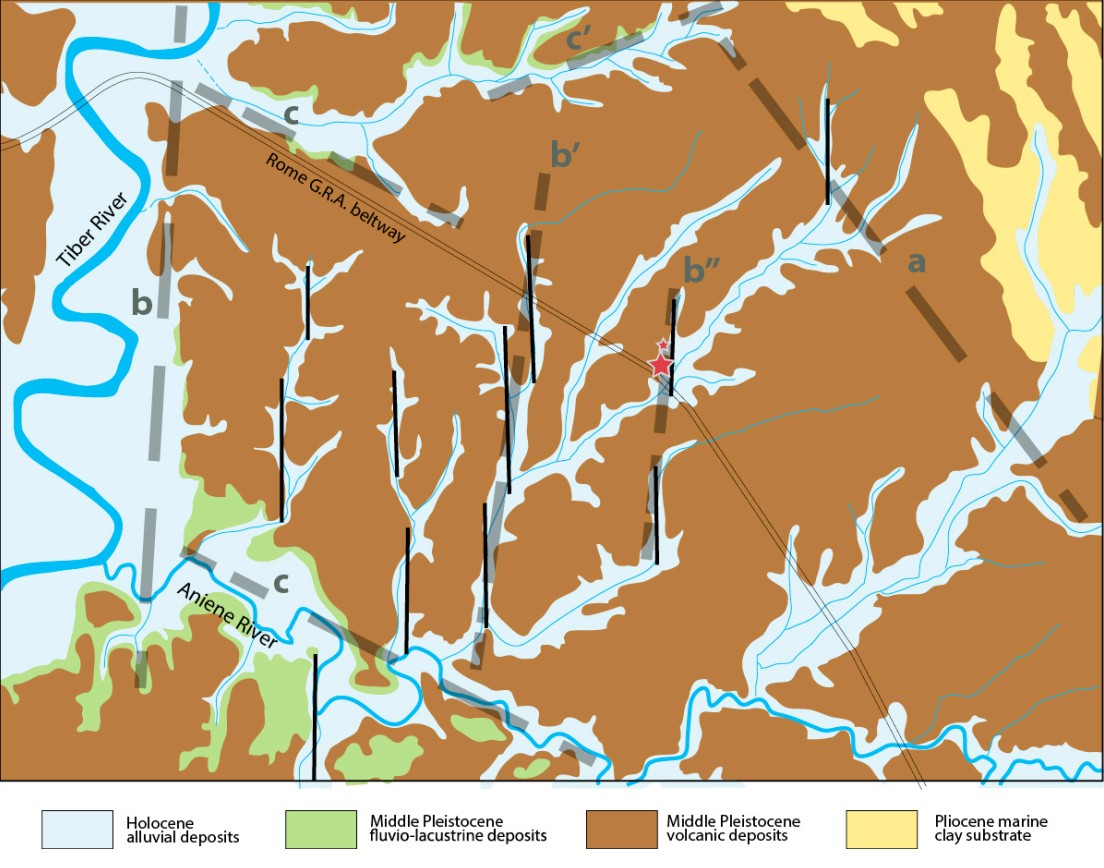

**Figure 89.** Geo-morpho-structural setting of the epicenter area. The thicker dashed lines
represent the main buried faults inferred from the analysis of the hydrographic network, with
the exception of the "a" fault, interpreted on the basis of the presence of a structural high to the
NE, represented by outcrops of Pliocene sediments. A fourth set of NE-SW lineaments is likely
originated by the topographic gradient in this area and is not highlighted as potential structural
control.  The thin, solid lines represent the superficial expression of the deformation linked to
faults that are continuous at depth (b', b"), evidenced by straight tracts of the riverbeds. One of
these deep NNE-SSW faults is the one that generated the May 11th earthquake, as the focal
mechanism of this event suggests.


ii- The second set of lineaments has a direction from NS to NNE-SSW ("b" in Fig.
9) and reflects that of even older faults, with right-lateral strike-slip character
(i.e. sub-vertical faults with right-hand horizontal movement (Alfonsi et al., 1991;
Faccenna et al., 1994). These faults are linked to the dismemberment of the
Apennine chain in independent arcs, due to the fragmentation of the "slab", that
is the "Adriatic" tectonic plate which subducted below the Apennine orogenetic
chain (Malinverno and Rayan, 1986; Patacca et al., 1990). However, these faults
have been active until recent times (Faccenna et al., 2008; Marra et al., 2004b),
probably due to the independent geodynamic mechanism that generated them,
and are competing with the regime of forces that originated the extensional
faults (Marra, 2001; Faccenna et al., 1996). We also know from the analysis of the
focal mechanisms of local earthquakes that small N-S fault segments are
currently reactivated with opposite movement (left-lateral) together with the
"Apennine", dip-slip faults (Frepoli et al., 2010).

iii- Finally, a third set of lineaments has conjugated WNW-ESE and ENE-WSW
directions ("c" and "c'" in Fig.9) and creates particular rhomboid "domains"[9].
Within these discrete regions, the N-S direction (as in the case of the epicenter
area of the Rome's May 11th, 2020 earthquake, Fig. 9), or the same WNW-ESE
directions (sectors 1A and 5A in Fig. 5B) may prevail. The origin of these
domains is linked to the strike-slip faults and can be generated between two
long, parallel N-S lineaments (Jones and Tanner, 1995). The characteristic of the
strike-slip (transcurrent) faults is precisely that of being arranged in parallel
with "en-echelon" geometry, that is, along stairway segments which can,
however, locally have a lateral overlap between them (Sylvester, 1988). The en-
echelon geometry characterizes the surface expression of faults that are
continuous at depth  (Sylvester, 1988) (examples b' and b" in Fig. 9).

**8. Conclusions**
The analysis of the hydrographic network in the epicenter area of the May 11th,
2020 earthquake shows a relative maximum concentration of the streambed in
the NNE-SSW direction: some of such rectilinear tracts, arranged with en-
echelon geometry, are highlighted in Fig. 5. We interpret these features as the
surface expression of buried NNE-SSW, strike-slip faults. Indeed, the focal
mechanism and aftershock alignment reveal that one of these buried ~N-S fault
reactivated with left-lateral movement on the occasion of the May 11th 2020
earthquake. Effectively, tectonically sensitive geomorphic analyses revealed the
occurrence of a cluster of knickpoints in the right side of the Aniene River that
can be classified as slope-break knickpoints and are aligned along NW-SO and N-
S orientation. Such a fluvial net perturbation corroborates the hypothesis of
recent tectonic activity affecting the study area along those faults.
When we consider the multitude of lineaments that are present at a wider and at
a smaller scale in this region (e.g., Fig. 2 and Fig. 9, respectively), we realize the
extreme fragmentation deriving from the intricate network of genetically
different faults. Such fragmentation results into a number of small fault
segments, with respect to the original long fault lines generated under the
competitive tectonic regimes that affected this region during Pleistocene times.
We remark that such high fragmentation is mainly provided by a en-echelon
system of ~N-S strike-slip faults which have crustal continuity. Therefore
hindering the lateral continuity of the NW-SE trending faults, which represent
the most favorably oriented fault system with respect to the Present-day NE-SW
extensional regime.
Small fault planes and a weaker tectonic regime explain the occurrence of
moderate seismicity and provide a likely explanation for the inhabitants of Rome
of the reason why they should not expect that a large earthquake may affect the
City.

**Additional information**
The authors declare no competing financial and non-financial interests.

**Data availability statement**
All data generated or analyzed during this study are included in this published
article.

**Author Contribution statement**
F.M. conceptualization, methodology, validation, investigation, Writing - Original
draft, supervision
A.F. methodology, validation, investigation, data curation, Writing - Original draft
D.G. methodology, validation, investigation, data curation, Writing - Original draft
M.S. methodology, validation, investigation, data curation, Writing - Original
draft
A.T. methodology, validation, investigation, data curation, Writing - Original draft
M.B. methodology, validation, investigation, data curation, Writing - Review and
editing
G.D.L. methodology, validation, investigation, data curation, Writing - Review and
editing
M.L. methodology, validation, investigation, data curation, Writing - Review and
editing

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
