# Peer review of "A morpho-tectonic approach to the study of earthquakes in Rome"

_Natural Hazards and Earth System Sciences, 2021_

## Author Comment (AC1)

[Figure]

[Figure]

**Figure 12** a-b) Interpretative geodynamic cross-section and (c) map of central Italy, explaining the superimposition of two competitive stress-field (extension and transpression) in this region (see text for explanations). Red stars are the hypocenters of the two earthquakes occurred in this region with fault plane solution showing anti-apenninic (NE-SW) P-axis.

---

## Author Response (AR1)

Dear Editor,
we are resubmitting a revised version of our paper in which we have addressed all the comments by the two reviewers, as detailed in the point-by-point answers.
The several modifications provided to the paper include:
i- Adjustments to the abstract and conclusions clarifying the implications and limits of the morpho-structural analysis on the assessment of the seismic hazard;
ii- A new dedicated section expanding the description of the tectonic feartures of the area of Rome, including a more informative version of Figure 2;
iii- An implemented section 2 on the seismicity of Rome, including a Table with the historical earthquakes that hit the City;
iv- Additional informationn of te seismic networks, the computation of the fiocal mechanism and the aftershocks, including a Table with the seismic parameters;
v- A new figure reporting the main results of the morphotectonic analysis of the drainage network.

We are confident that these significant modifications have fulfilled the requests by the reviewers and strongly improved the clarity and the quality of our paper.

Thank you for your kind attention,
Best Regards,
Fabrizio Marra and co-authors

REVIEWER #1

The paper presents an attempt to infer the hidden seismotectonic setting from a morphological analysis. In this regard, the study may be of some interest to improve the knowledge of the Rome area. Beyond this possible contribution, no inference about the present seismic hazard can be obtained as claimed by the Authors in the astract and conclusions.
In my view, by no way the analysis carried out supports this statement: neither as concerns the "weakness" of the present tectonic regime
nor the small dimensions of the faults (which, as expression of deeper fault systems that may be not segmented as the respective surface expressions seem to be).
In essence the main conclusion (low hazard) only relies on the lack of strong earthquakes in the historical records (which is a strong evidence in my view). Thus, abstract and conclusions should be modified to reduce ambitions of the paper.

While we agree that we may have overestimated the implications of our study on the assessment of the hazard for the city of Rome, we would like to clarify some issues in order to defend, at least in part, and to support further our assumptions.

As reported in the abstract and detailed in the section dedicated to the seismotectonic features of the Rome's area, the longest historical record in the world attests that no destructive earthquake affected the city of Rome during the last 2000 years. We agree with the Reviewer that this fact alone may be indicative of a low hazard. However, aim of our study is to provide a possible explanation (and we agree that we are not providing THE explanation) for this moderate seismotectonic regime.

Therefore, while we agree that abstract and conclusions should be modified to smooth the direct implications of the morpho-structural analysis on the assessment of the seismic hazard, we believe that the characterization of the surface expression of the active faults in this area provides inferences on the interpretation of the present seismic hazard. We agree that such inferences could not be substantiated without combining the morpho-structural study with that of the historical and instrumental seismicity, but this is what we actually were intended to do.

According to these considerations, we have modified the conclusive sentence in the abstract as follows:

*"Small faults and a present-day weaker tectonic regime with respect to that acting during the Pleistocene might explain the lack of strong seismicity in the long historical record, suggesting that a large earthquake is not likely to occur."*

Regarding the possibility that the small surface faults may be expression of a deeper fault systems that may be not segmented as the respective surface expressions seem to be, we have added the following sentence in the conclusion paragraph:

*"We remark that such high fragmentation is provided by a en-echelon system of ~N-S strike-slip faults which have crustal continuity. Therefore hindering the lateral continuity of the NW-SE trending faults, which represent the most favorably oriented fault system with respect to the Present-day NE-SW extensional regime. "*

As concerns the seismotectonic interpretation, it is not clear to me what is the origin of the new "competing" tectonic regime responsible for the sinistral reactivation of dextral strike slip faults. Possibly this is not the core of the present paper: in this case, discussions about active strain regimes could be safely removed by only focusing the paper on inferring the apparent geometry of fault systems.

We have added a new section describing more in detail the tectonic features of the area.

REVIEWER #2
ANSWERS TO REVIEWER #2's COMMENTS

1.The general frame depicted in Section 2 would benefit from a wider description of the seismic events that hit Rome in the past, also including those occurred in "the period of ancient Rome, as well in the Early Middle Ages" (lines 84-86). This information can be easily retrieved in the available seismic

catalogues and, in general, in the literature. This would show that comparable damage (e.g., intensity VI-VII) has been caused both by strong earthquakes with a far epicentre, e.g., in the Apennines chain, and by moderate events much closer. For this reason, defining the potential size of these moderate events significantly contributes to a better definition of the seismic hazard of the area.

I suggest also adding a new figure with a graph or a table representing the seismic history of Rome. In the same line, it would be useful to know the magnitude of the instrumental earthquakes. Are there any comparable with the 2020 Ml 3.3 event? In case, they could be outlined in Figure 1. Concerning this figure, there are also some details that need to be fixed: the blue star of the 2020 event is not so evident; there are letters A-B and C-D that are not defined in the caption; in the legend, Bulletin has two t; neither in the caption nor in the text a definition of G.R.A. is reported (only in Figure 8, at the end of the paper). Could you add the stream of the Aniene river? It would help compare this one with the other figures.

1. We have implemented section 2 according to the suggestions by Reviewer #2. However, the earthquakes dated to the ancient periods of Rome history are very poorly constrained. In fact they are based on one account only, and directly referred to the city of Rome. This fact does not allow us to distinguish far earthquakes from close events. During Roman Ages, the definition of the term "Rome" was often associated with entire territorial possessions, e.g. the whole Italy.

To better illustrate this topic we have added the suggested table with the earthquakes that hit Rome with Intensity greater than 6.

We have provided all the requested modifications and corrections in Figure 1 and in its caption.

2.The description of the structural setting could be more precise, even without being longer, and for this purpose an improved Figure 2 would be of great help. In general, this figure needs to be rethought for an international audience unfamiliar with the tectonics of Central Italy. Which is the age of the tectonic features reported (thrusts, normal and strike-slip faults)? Which are active today and which not? Why the extensional stress field in the Apennines has a different graphic than that along the Tyrrhenian margin? Is the stress field of the strike-slip faults no longer active (the retrieved focal mechanism has an opposite kinematics)? In the text, you talk about the volcanoes of the "Roman Province", whereas in this figure you represent the volcanic districts of the Tyrrhenian Sea margin. Could you homogenize the names, also highlighting the Colli Albani? Could you add a box corresponding to figure 1 and a graphic scale? Could you add any references in the figure caption?

I suggest redrafting Figure 2 and then rewriting coherently the structural setting.

2. We have improved Figure 2 adding the requested information on the age of the tectonic features and on the different extensional stresses, along with the other suggested modifications. Moreover, in order to accomplish also to the request by Reviewer #1, we have introduced a new section to describe more in detail the tectonic features of the investigated area.

Concerning the morphological setting, Figure 3 is not centred on Rome and does not include the 2020 Ml 3.3. I suggest reframing the figure, expanding it to the North and to the West.

We have slightly enlarged the area in Figure 3 in order to include the epicenter of the 2020 seismic event.

3.The way the seismicity is addressed in the paper should be better organized. The seismicity of the area is described in Section 2. Section 4, called Seismicity, describes the data collection, but it also shows a part of methodological description, in particular the hypocentres relocation. It does not mention, however, the computation of the moment tensor solution, that is addressed directly in the Results, but is shown in Figure 4, to which Section 4 refers. Now it seems that the focal mechanism in Figure 4 comes from the literature.

3. We reamark that there is no moment tensor computation; we have computed a focal mechanism of the mainshock using first-motion polarities (57 P-wave polarities) with the code FPFIT (Reasenberg and Oppenheimer, 1985). Focal mechanism with first-motion polarities is shown in figure 4.

We have provided a revised Fig. 4 in which we have reported all the four arrays with different colors. In the map are shown only the nearest seismic stations with respect to the epicenters.  The 4 arrays are extended in the whole Central Appennine and are all used in the relocation of the small sequence.

I also suggest adding a table with the list and the parameters of the Ml 3.3 event and the 4 aftershocks (magnitude, depth, etc.) mentioned at page 14.

We have also provided a table with the list and the parameters of the Ml 3.3 event and the 4 aftershocks.

Why only 2 out of these aftershocks are shown in Figure 4?

We originally preferred to not add the location of two mainshocks because they are localized far from the mainshock's epicenter. However, we have now added them in the revised Figure 4b and we have discussed in the text the fact that they are not related with the seismogenic structure.

This example highlights, as a more general point, the need of a clearer organization of the text. I suggest reviewing the Table of Contents of the paper, separating better the introductory framework, the data used, the methodologies adopted, and then results and discussion. Within each of these general topics, subsections regarding the different disciplines (seismology, geomorphology, tectonics, etc.) need to be included. Otherwise, as it happens now, you have a mix of literature, data and interpretations in many parts of the paper, and this does not help the reader.

We have inserted a new heading to separate the introductory paragraphs from those describing data and methods.

4.The morphotectonic analysis of the drainage network (Section 6.3) is a huge work, supported by several detailed figures. However, the tectonic lineaments that are present in all these figures do not allow a proper view of the data analysis, whose details are masked by the black lines. Therefore, on the one hand, I suggest removing the tectonic lineaments from Figure 6 a) and b) and Figure 7.

On the other hand, the Authors should add a new figure where all the main results from the previous figures are reported (fluvial elbows, knickpoints, etc.) along with the interpreted tectonic lineaments.

5. We have removed the tectonic lineaments from Figure 6 a) and b) and Figure 7 and added a new figure where all the main results from the previous figures are reported (fluvial elbows, knickpoints, etc.) along with the interpreted tectonic lineaments.

Moreover, it is not clear why the Authors draw a N-S lineament near the Ml 3.3 epicentre even though in figure 5 A) it falls within a zone characterised by NE-SW streambed analysis (see also lines 346-349 at page 15).

We remark that the NE-SW directions are clearly overprinted by  the N-S ones in this area. The focal mechanism of the event clearly indicates that it occurred on a conjugated set of N-S and E-W planes, therefore excluding a NW-SE structure from being the fault plane. Such elements to identify the surface lineament associated with the seismogenic structure at depth are clearly exposed in the paper.

5.The most critical point, in my view, is in the concept of "seismic intensity" that, according to the Authors, the analysed faults have now compared with that they had in the geological past. This

concept, along with the seismic intensity of the area related to the Pleistocene stress field, is present since the beginning of the paper (page 7, from line 153), up to the Discussion.

I think that this concept should be totally revised. In general, seismogenic faults are not characterised by a "seismic intensity" but, rather, by a "seismic rate", that can be estimated if you are able to recognise one or more seismic events that they generated in the past (for instance from historical and/or palaeoseismological record), associated with a "geological slip rate" from structural, geomorphological and stratigraphic data.

In this study, the faults analysed are buried and blind, there are only hints of their activity at surface. Therefore, there is no information to assess which is their current and past activity and seismic rate. Moreover, there are no data to discuss the "dimension" of the current and previous stress fields, although it is clear that the Middle Pleistocene tectonic activity, also responsible for the development of the volcanic district, was much more developed than the current tectonic activity.

I think that a scheme or a table is needed reporting (with refs) the orientation, kinematics and age of the different stress fields (including the strike-slip one of figure 2) that affected the study area through time and that are relevant to this study. Based on this, the inception, development, segmentation and possible reactivation of faults can be framed and discussed. This will allow the Authors to strengthen their idea that segmented faults with limited tectonic activity can be assigned a seismogenic potential for events with moderate magnitude.

6. The Reviewer highlights the core of the problematics concerning the assessment of the seismic risk for Rome: in this area the seismogenic structures are buried and blind, and there are only hints of their activity at surface. Therefore, there is no information to assess which is their current and past activity and seismic rate. We believe that one indirect way to provide an estimation of the seismic potential is the morpho-structural aproach, aimed at providing information of the size of the potential faults.

On this regard, we have implemented Figure 2 also with a crustal cross-section which approaches the scheme suggested by Reviewer #2, reporting the orientation, kinematics and superposition of the different stress fields that affected the study area since Messinian to Present time.

**Details**

Page 3, lines 79-81

I suggest describing what the Greater Rome is: the Province of Rome?

Yes. We have specified it.

Page 6, lines from 139

The name of the volcanic complex should be added.

Done.

Page 7, lines 148-151

This part should be described better, and some references added.

OK.

Page 7, lines 157-159

This sentence (Moderate earthquakes … almost exclusively … in the volcanic area) is not supported by the current figure 1, where moderate earthquakes are reported also in correspondence with the city. Maybe, the modified Figure 1 could help clarify this part.

Almost exclusively doesn't mean that no moderate earthquaqke occurred in Rome. We have modified Fig. 1.

Page 7, line 170

"At depth": which depth? Could you characterize better the third dimension of these faults?

We have added a ccross-section in Figure 2 that visualizes it.

Page 8, line 186

The Italian Strong Motion Network (RAN) should be mentioned along with its owner/operator, as you did for RSN and INGV, RSA and Lazio and Abruzzo regions, IESN as an amateur seismic network. RAN is operated by the National Civil Protection Department.

OK.

Page 10, line 206

"Carried out" rather than "carried on".

OK.

Page 11, Figure 5 B)

Add in the caption what the light blue and yellow lines are.

OK.

Page 12, lines 240-249

Here the Authors could explain the reason why a role played by lithologies can be ruled out. This concept, without explanation, can be read at page 16, line 367. Maybe a figure with a geological sketch could be added.

Geology is reported in Figure 9. We have highlighted in the text thyat the drainage network affects a heterogeneous geologic substrate,

Page 14, lines 296-306

At page 7, line 170, the Authors state that the fault planes at depth do not propagate to the surface. Therefore, here some explanation is needed on the mechanism causing fault induced disturbance on some elements of the drainage network.

We have clarified it.

Page 14, line 325

I strongly discourage the use, here and in other parts of the text, of the term "Antiapennine" as a synonym of a NE-SW direction, and of the term "Apennine" as a synonym of a NW-SE direction. Please, refer to NE-SW, etc.

"Antiapenninic" is use only once, followed by (NW-SE).

Page 16, line 363

Check the number of the figure.

OK.

Page 16, line 375

Check the orientation.

OK.

Page 20, line 396

Remove "by the INGV": references are already there.

OK.